# Multichannel input pixelwise regression 3D U-Nets for medical image estimation with 3 applications in brain MRI

**Jueqi Wang**[1]          X2019CWN@STFX.CA **Derek Berger**[1]          DBERGER@STFX.CA

**David Mattie**[1]          DMATTIE@STFX.CA **Jacob Levman**[1]          JLEVMAN@STFX.CA

[1] *Department of Computer Science, St Francis Xavier University, Canada*

## Abstract

The U-Net is a robust general-purpose deep learning architecture designed for semantic segmentation of medical images, and has been extended to 3D for volumetric applications such as magnetic resonance imaging (MRI) of the human brain. An adaptation of the U-Net to output pixelwise regression values, instead of class labels, based on multichannel input data, has been developed in the remote sensing satellite imaging research domain. The pixelwise regression U-Net has only received limited consideration as a deep learning architecture in medical imaging for the image estimation/synthesis problem, and the limited work so far did not consider the application of 3D multichannel inputs. In this paper, we propose the use of the multichannel input pixelwise regression 3D U-Net (rUNet) for estimation of medical images. Our findings demonstrate that this approach is robust and versatile and can be applied to predicting a pending MRI examination of patients with Alzheimer's disease based on previous rounds of imaging, can perform medical image reconstruction (parametric mapping) in diffusion MRI, and can be applied to the estimation of one type of MRI examination from a collection of other types. Results demonstrate that the rUNet represents a single deep learning architecture capable of solving a variety of image estimation problems. Public domain code is provided[1].

**Keywords:** 3D U-Net, Image Synthesis, Medical Image Reconstruction, Deep Learning

## 1. Introduction

The U-Net (Ronneberger et al., 2015) is a robust deep learning architecture designed for semantic segmentation in medical imaging that has been extended to 3D (Milletari et al., 2016). In this paper, we apply the pixel-wise regression U-Net to medical image estimation/synthesis, which has been previously applied to satellite images (Yao et al., 2018), and was given limited consideration in a study on intensity normalization in 2D (Reinhold et al., 2019). We propose the use of multichannel input 3D U-Nets (rUNet) that utilize multiple input volumes for prediction in three tasks: (1) Longitudinal image estimation in Alzheimer's, (2) Diffusion image reconstruction through learned parametric mapping, (3) MRI cross-modality estimation. Alternative methods using generalized adversarial networks and custom designed deep learning architectures have been previously proposed for image estimation of medical images (Yang et al. 2020, Zhou et al. 2020, Chartsias et al. 2018). These generally represent architectures that are more challenging to implement when compared to the relatively small adaptations necessary of the off-the-shelf U-Net in order for the architecture to robustly handle the image estimation problem. We also compare multichannel to single channel input.

---

[1] https://github.com/stfxecutables/Multichannel-input-pixelwise-regression-u-nets

## 2. Methods

**The U-Net model.** We use a 5-level 3D U-Net architecture[2], modified from[3] with Leaky ReLU activation ($\alpha = 0.2$), learning rate ($\alpha = 10^{-5}$), Adam optimizer, mean absolute average error (MAE) loss function, z-score intensity normalization, early stopping which stops training when no improvement is observed at validation loss, and co-registered volumes resized to $128 \times 128 \times 128$. Applications 1 and 3 included skull stripping. Batch size was 3 in applications 1 and 3, and 1 in application 2. We compare all approaches with mean squared error (MSE), MAE, structural similarity index measure (SSIM), and peak signal to noise ratio (PSNR). **1. Longitudinal image estimation.** Data was used from the Alzheimer's Disease (AD) Neuroimaging Initiative database[4]. 88 AD subjects with 4 longitudinal MRI scans at screening (SC), month 6 (M06), month 12 (M12) and month 24 (M24) were used, with 70 subjects included for training and 18 for testing. We evaluated using all 3 preceding scans (SC, M06, M12), 2 scans (M06, M12), and only one scan (M12) to predict the M24 volume. **2. Diffusion image reconstruction.** Data was obtained from The Human Connectome project[5], where we used 19 MR diffusion tensor exams randomly divided: 16 training, 3 testing. The Diffusion toolkit[6] was used to process the 4D diffusion exam into fractional anisotropy (FA) and apparent diffusion coefficient (ADC) volumes for the target images. Data augmentation was employed, including rotation with random angle $[-12°, 12°]$ and a random spatial scaling factor $[0.9, 1.1]$. **3. T1/T2 estimation.** Data was obtained from BraTS'18-'20[7], containing T1, T2 and T2-FLAIR volumes from 1044 training and 261 testing subjects. We have tested estimating T2 from T1 and from T1 & T2-FLAIR, and estimating T1 from T2 and from T2 & T2-FLAIR.

## 3. Results and Discussion

Quantitative results (Table 1) and example predicted images (Figure 1) are provided. Results demonstrate improvements in MSE, MAE, SSIM and PSNR from the use of multichannel input relative to single channel input in applications 1 & 3. Multichannel input is required for application 2. Diffusion parametric mapping demonstrated that, similar to the regular U-Net, we get robust results with very small datasets when employing data augmentation. Results indicate that the 3D multichannel rUNet is a robust and flexible architecture capable of handling a diverse array of image estimation problems, with minimal reconfiguration/retuning, while achieving strong performance. The AD application predicts future rounds of imaging, potentially providing useful information for clinicians in charge of managing a patient's care and monitoring neural degeneration. The diffusion application demonstrates potential for the rUNet in image reconstruction applications, supporting the creation of learned parametric maps that can target any given spatially distributed anatomical or physiological measurement of interest. The diffusion and T1/T2 applications demonstrate the potential for performing image reconstruction/estimation nearly instantly (prediction time is 0.2s on a GPU), thus having potential in emergency and critical care circumstances where near instant reconstruction/estimation for clinicians is valuable, and situations where there is value in limiting the patient's time in the MRI scanner.

---

2 https://github.com/stfxecutables/Multichannel-input-pixelwise-regression-u-nets
3 https://github.com/fepegar/unet
4 http://adni.loni.usc.edu/
5 https://www.humanconnectome.org/
6 http://trackvis.org/dtk/
7 https://www.med.upenn.edu/cbica/brats2020/

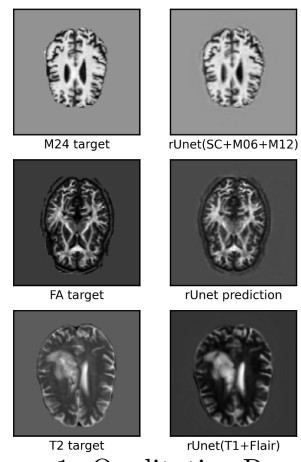

M24 target | rUnet(SC+M06+M12)

FA target | rUnet prediction

T2 target | rUnet(T1+Flair)

Figure 1: Qualitative Results.

Table 1: Quantitative Results.

| Task | MSE ↓ | MAE ↓ | SSIM ↑ | PSNR ↑ |
|---|---|---|---|---|
| SC+M06+M12→M24 | 0.0058 | 1.952 | 0.967 | **35.056** |
| M06+M12→M24 | **0.0056** | **1.269** | **0.968** | 34.937 |
| M12→M24 | 0.0065 | 1.756 | 0.961 | 34.292 |
| Diffusion→ADC | 0.0253 | 1.239 | 0.919 | 35.423 |
| Diffusion→FA | 0.0136 | 1.258 | 0.945 | 35.980 |
| T1+T2-Flair→T2 | **0.0043** | 2.211 | 0.985 | **42.530** |
| T1→T2 | 0.0049 | **2.040** | **0.986** | 41.780 |
| T2+T2-Flair→T1 | **0.0030** | **0.900** | **0.989** | **42.604** |
| T2→T1 | 0.0041 | 1.137 | 0.986 | 40.872 |

## Acknowledgments

This work was supported through a Canada Research Chair grant (231266), a CFI and NSRIT grant, and an NSERC Discovery Grant to JL. Data collection and sharing was funded by the Alzheimer's Disease Neuroimaging Initiative (ADNI) (NIH Grant U01 AG024904) and DOD ADNI (W81XWH-12-2-0012).

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
