# OpenReview forum: "Multichannel input pixelwise regression 3D U-Nets for medical image estimation with 3 applications in brain MRI"
_MIDL.io/2021/Conference/Short — MIDL 2021 Poster_

### Official Review · Reviewer_gS1v · 2021-04-26

**Confidence:** 3
**Final Rating:** 3

**Summary:**

The authors propose to use pixel-wise regression 3D U-Net with multichannel input (3D multichannel rUNet) to solve image estimation problems. They apply their approach to three applications: (1) Longitudinal image estimation (2) Diffusion image reconstruction (3) MRI cross-modality estimation. The results indicate their proposed method can achieve reasonable performance in all three applications. The use of multi-channel input can improve performance in the application (1) and the application (3).

**Strengths:**

1. The combination of the 3D rUNet and the multichannel input method seems to be novel.
2. The evaluation results from the three applications are promising. The proposed method has the potential to solve a variety of image estimation problems.


**Weaknesses:**

1. Lack of comparison with other state-of-the-art approaches. The proposed method is tested on public datasets. Further comparison with other methods can improve the results section of the paper.
2. Consider the dataset size for application (1) and application (2) is small. The authors did not apply cross-validation for these applications.

**Deanonymize Review:**

yes

**Justification Of The Rating:**

The method proposed by the authors has some novelty. The results from the three applications prove that the proposed method can solve a range of image estimation problems. Therefore, I recommend the acceptance of the paper.

**Paper Type:**

both

**Special Issue:**

no

---

### Official Review · Reviewer_VuYT · 2021-05-01

**Confidence:** 4
**Final Rating:** 3

**Summary:**

This paper validated the use of 3D U-Net with multi-channel input for image-to-image mapping tasks in MRI. The motivation of this work was to demonstrate the flexibility and robustness of multi-channel 3D U-Net for various image estimation problems. Experimental results showed consistently better performance of multi-channel input than single-channel input on three tasks.

**Strengths:**

The claim that multi-channel 3D U-Net was versatile for various image estimation tasks was proved with extensive experiments, which was the main strength of the paper. Besides, method description was informative regarding different datasets used in the experiments.

**Weaknesses:**

1. Brain maps in Figure 1 were flipped up-and-down. Please fixed it.
2.  It will be more interesting to map T2FLAIR from T1 and T2 weighted images as the acquisition time of T2FLAIR is usually longer than T1w and T2w.

**Deanonymize Review:**

no

**Justification Of The Rating:**

This is a good validation paper to demonstrate the effectiveness of multi-channel 3D U-Net for various image-to-image mapping problems in MRI. Some Figure display issue should be fixed and additional T2FLAIR mapping experiments would be interesting if added.

**Paper Type:**

validation/application paper

**Special Issue:**

no

---

### Meta-Review · Area_Chair_CQdt · 2021-05-09

**Recommendation:** Accept (Poster)
**Confidence:** 3

**Metareview:**

Both reviewers give a weak accept for this interesting work on multichannel reconstruction, they do however mention missing comparisons to related work. While space is limited, we would strongly urge authors to include those into the GitHub repository for improved impact.

---

### Decision · Program_Chairs · 2021-05-11

Accept (Poster)